# Impacts of Learning Orientation on the Modeling of Programming Using Feature Selection and XGBOOST: A Gender-Focused Analysis

HoSung Woo [1]  and Ja-Mee Kim [2],*

1   Department of e-Learning Graduate School, Korea National Open University, Seoul 03087, Korea; hughwoo@knou.ac.kr
2   Major of Computer Science Education, Graduate School of Education, Korea University, Seoul 02841, Korea
*   Correspondence: celine@korea.ac.kr

**Abstract:** In programming, modeling is a generalized explanatory system that organizes key characteristics of a specific matter or object of interest using computer science concepts. Modeling is integral for both automation design in programming education and communication in the collaborative process. This study investigates the effect of learning orientation on the modeling stage based on gender. The study includes 756 male and 688 female elementary-school students. We analyzed the results of XGBOOST by extracting the influential characteristics from feature selection along with the basic statistics. As a result of the study, it was confirmed that learners, regardless of gender, had the largest gap in modeling and that this was the stage at which differences occurred in programming education. For male students, the process of collecting data for modeling or devising a solution was found to be an important learning method. This shows that it is necessary to create an environment to focus on activities that derive solutions from the collected data along with strengthening information retrieval education. Although female students showed a similar tendency to male students, the process of cooperating with friends as a differentiating factor was found to be an important learning method. It seems necessary to apply teaching and learning methods that can strengthen team projects that can collaborate with friends. The findings could serve as a reference for teaching and learning design and operation for effective programming education.

**Keywords:** learning orientation; programming; gender; feature selection; XGBOOST

## 1. Introduction

At the national level, gender balance is one of the most important goals for achieving national prosperity and a fair and responsible social model [1]. Initially, the gender balance was not a matter of concern in computer science [2]. However, the gender gap in computer science is increasingly widening with the advancement of the information technology (IT) industry [3,4]. Despite the increased social participation of women, the economic participation rate of Korean women is still below the Organization for Economic Co-operation and Development (OECD) average, and the proportion of women in Software (SW) related occupations was low at less than 15.0% between 2011 and 2015 [5–7]. No gender distinction must exist in computer science, and in any area for that matter, and the utilization of women as a national resource must be considered [8].

Learning is the change in an organism as a result of an experience, which, in turn, influences its behavior. Learners accumulate new experiences and knowledge through learning, based on which an individual develops and adapts [9]. With a growing emphasis on computer science education, an increasing number of studies have analyzed gender differences in learning, in the areas of computational thinking, interest in programming, unplugged, and career in computer science [10–14]. If gender differences do exist, it would

be necessary to provide appropriate and complementing educational services to help learners have a meaningful learning experience.

Programming is the ability to process and solve problems based on various knowledge and experiences using computing power. According to a substantial body of research, the gender of learners is associated with academic achievement as a predicting variable [10–15]. It has been reported that boys are traditionally more familiar with technology and take less time than girls to reach a particular learning level [16,17]. Moreover, it is argued that boys are generally more experienced with computers, more engaged, and have a more positive attitude toward computers compared to girls [18,19]. In contrast, opinions have been shared that the gender of learners does not influence learning in the computer science field [20–24]. Thus, the findings on gender differences in the literature are mixed. It will be difficult to conclude or interpret the analysis results simply as gender differences, as there are differences in complex factors, such as school level or grade level, class content, teaching method, and learner's learning orientation.

Against this backdrop, this study aimed to analyze the differences in programming modeling and learning orientation based on gender. To this end, a statistical analysis technique was adapted to extract the influential features in modeling, which was verified with an ensemble-based prediction model that showed excellent performance in data science competitions.

## 2. Research Background

### 2.1. Learning Orientation and Programming

Learning orientation implies that the most effective teaching or learning method differs with the individual. This can be defined as the characteristic cognitive, emotional, and psychological behaviors of how learners perceive, interact with, and respond to their learning environment [25].

Some researchers argue that learning orientation is a useful indicator of potential academic success, as it provides information on individual differences in learning preference and problem-solving capabilities [26,27]. Learning orientation represents how the learner learns and prefers to learn, and helps instructors adapt to individual student needs in order to teach them successfully [28,29].

A previous study examined the relationship between Myers–Briggs personality type and computer programming. According to the results, sensing learners exhibited better results than intuitive learners on programming tasks, with a higher average score on programming tasks [30]. However, no significant differences were observed between all personality types and final programming scores. In a follow-up study, the learning orientation toward the programming stage was analyzed [31].

A study investigated the relationship between learning orientation (independent/dependent) and the academic achievement of students enrolled in computer courses [32]. The academic achievement was measured by summing up the scores of general computer use, basic programming, and computer knowledge. Independent students scored much higher than dependent students. The difference was much more evident in basic programming and computer knowledge than in general computer use. According to this study, learning orientation influences programming, and those with a specific type of learning orientation perform better in a specific programming environment [32].

Since this study targets elementary school students, it is highly relevant to beginners or novices in programming. According to a study that examined the effect of learning orientation achievement in the introductory programming class [33], reflective learners received better grades than active learners, and verbal learners performed better than visual learners.

Taken together, the above-mentioned studies show that the learner's personality type is not related to achievement, but the learning disposition is related to academic achievement. It seems to be meaningful to determine the type of characteristics involved in the modeling process of programming according to the learner's learning propensity [34–36].

### 2.2. Gender and Programming

Problem solving in computer science presupposes the use of computing systems. Solving problems with the computing system is closely related to programming education, as far as automation is considered [37].

The learning effect of programming education has been found to be different across genders [38]. Studies around the 1990s argue that men are generally more experienced with computers and are more capable of using programming languages in a more complicated way [39,40]. Since computer anxiety has a negative correlation with the use of a computer, the more experienced the user becomes, the lesser their level of anxiety [41]. This result is in the same vein as the finding that men tend to have lower anxiety about using computers and exhibit more interest in them as compared to women [42,43].

A 2010 study found that male students showed higher achievement in using computing tools, such as programming [44], while female students exhibited a lower flow level in programming than male students [45]. It was reported that female students were more interested in theories, demonstrating higher achievement in this domain, and that basic education should be emphasized to enable them to apply the basic knowledge into practice. Meanwhile, male students were reported to be more interested in the practical aspect, in which they had higher achievements [5]. Further, in robot programming learning, female students exhibited lower motivation, interest, and attitude toward learning programing than their male counterparts, with higher cognitive pressure regarding robot programming learning [45].

In contrast, some studies have reported no gender differences in programming education [23,24]. For example, in the introductory Python programming class, the students' attitude toward programming was measured [19]. Attitude is an ambiguous factor to measure, but it was approached based on four aspects: confidence in computer programming learning, usefulness, attitude toward success, and effective motivation. The average score of attitude was significantly higher among boys than girls. Regarding confidence in programming and the usefulness of programming and motivation, male students scored higher on average than female students. Another study noted differences in sub-factors, although programming does not show a big gender difference [16]. After learning a computer science program on computer memory concepts, students were asked to fill out the computer memory knowledge test (CMKT) questionnaire to measure the effectiveness of learning. Although there were no gender differences in the post-test scores, significant gender differences were observed in accessibility to learning materials and the three sub-factors, but not in the other sub-factors.

In summation, the research findings in the literature on gender differences in computer science education have been mixed. Research on the effects of gender on computer programming performance is extremely complicated and may produce inconclusive results [46]. Nonetheless, amid the growing national interest in IT, an understanding of gender differences in programming education will help overcome the effect on programming education weakening. Thus, it will greatly help in setting educational directions to closely examine the gender difference in programming education.

### 3. Method

In this study, we developed two tools to analyze the effect of learning orientation in the modeling of programming. The first tool is the SW competency assessment tool. It was developed based on four steps: analysis, modeling, implementation, and generalization. These are the problem-solving processes of programming. The other tool is a questionnaire to understand the learning propensity of learners. The tool that was used in the Japanese information literacy survey was used.

For the analysis method, a combination of statistical approach and machine learning technique was used. In other words, before applying the XGBOOST, a regression model-based feature selection was applied. In the data analysis process, selecting data with similar characteristics or extracting data with important characteristics greatly affects the research

results [47]. Because feature selection is an effective technique for extracting data with important features, we used both analysis methods together. The specific details are as follows.

### 3.1. SW Competency Assessment Tool (CAT)

We conducted a literature review on SW competency to develop an SW competency assessment tool (CAT). As a result of the literature analysis, Korea's computational thinking-based problem-solving ability evaluation questions emphasize the problem-solving process, and SW competency is divided into analysis ability, design ability, implementation ability, and reasoning ability. Lithuania's Bebras Computing Challenge is not just a diagnostic tool for evaluation but consists of questions that support the understanding of computer science while solving problems. Japan's information literacy assessment tool includes computer science and ethical content in terms of practical ability to use information, scientific understanding of information, and an attitude to participate in the information society. Germany's Abitur is a graduation exam and focuses on evaluating professional knowledge in computer science. Based on our findings, we set the direction for SW competency and tool development. To review the suitability of the SW competency assessment tool, three times of expert councils and pilot tests were conducted for 2 professors of the computer area and 10 teachers to revise and supplement the questionnaire items to suit the purpose of the study. Research procedure details are summarized in Table 1.

**Table 1.** SW CAT development procedure.

| Step | Div. | Description |
|---|---|---|
| 1 | SW CAT Analysis | Derive SW competency and set the tool development directions<br>- Computational thinking-based problem-solving ability (ROK, Confidential)<br>- Bebras Computing Challenge<br>(Lithuania, www.bebraschallenge.org/, [accessed on 9 May 2022])<br>- Information literacy<br>(Japan, www.mext.go.jp/, [accessed on 9 May 2022])<br>- Abitur<br>(Germany, www.standardsicherung.schulministerium.nrw.de, [accessed on 9 May 2022]) |
| 2 | First Expert Review | Two professors (one in computer science and one in computer education)<br>Review the derived SW competency<br>Review tool development direction |
| 3 | SW CAT Development | Develop questions considering the first expert review |
| 4 | Second Expert Review | Two professors (one in computer science and one in computer education)<br>Review the developed items |
| 5 | Revision and Supplementation of Questions | Revise and supplement the questions considering the second expert review |
| 6 | Conduct a Pilot Test | Test conducted on 72 students from two elementary schools |
| 7 | Third Expert Review | Teachers, by school level (four elementary school teachers, two middle school teachers, and four high school teachers)<br>Review the composition, expression, and difficulty of the questions |
| 8 | Pilot Test Result Analysis | Group score analysis<br>Question analysis |
| 9 | Revision and Supplementation of Questions | Revision/supplementation of questions based on pilot test results and content review by field teachers |
| 10 | Development Completed | SW CAT development completed |

The total number of items in the developed SW CAT is 17, and the difficulty and degree of discrimination are shown in Table 2.

**Table 2.** Test results for elementary school.

| SW Competency | Question No. | Difficulty (0.62) | Discrimination (0.462) |
|---|---|---|---|
| 1. Analysis | 1.1 Data Collection | 0.35 | 0.404 |
| | 1.2 Data Analysis | 0.34 | 0.412 |
| | 1.3 Data Collection | 0.84 | 0.389 |
| | 1.4 Data representation | 0.78 | 0.270 |
| | 1.5 Data representation | 0.55 | 0.421 |
| | 1.6 Data Analysis | 0.73 | 0.483 |
| 2. Modeling | 2.1 Problem Decomposition | 0.36 | 0.274 |
| | 2.2 Algorithms | 0.70 | 0.588 |
| | 2.3 Algorithms | 0.73 | 0.549 |
| | 2.4 Abstraction | 0.52 | 0.420 |
| | 2.5 Problem Decomposition | 0.51 | 0.510 |
| | 2.6 Algorithms | 0.57 | 0.481 |
| 3. Implementation | 3.1 Automation | 0.15 | 0.250 |
| | 3.2 Automation | 0.13 | 0.252 |
| | 3.3 Testing | 0.48 | 0.477 |
| | 3.4 Testing | 0.36 | 0.517 |
| 4. Generalization | 4.4 Application and generalization | 0.53 | 0.484 |

The test tool consists of six analysis questions (35.3%), six modeling questions (35.3%), four implementation questions (23.5%), and one generalization question (5.9%). Each SW competency includes elements of data collection, data analysis, data representation, problem decomposition, abstraction, algorithm, automation, testing, application, and generalization. Each factor considered the computational thinking-based problem-solving process.

Since the generalization questions were considered inappropriate due to the difficulty level for elementary school students, only one question was implemented after expert review. The average of item difficulty and discrimination was 0.62 and 0.462, respectively, which were evaluated to be excellent [36].

The SW competency is defined as follows. Analysis is a factor that can present necessary information in various types of data such that they can be used. Modeling, as a process of expressing complex problems in an easy-to-use format for a purpose, is a factor that reduces complexity by breaking a problem into smaller tasks that can be easily addressed, represented as a sequence of procedural steps. It is an important factor in designing an algorithm. Implementation enables the planned and modeled items to be realistically operated in the system. Generalization is a factor that expands the scope of the structured model to include more cases. This factor allows expansion to a wider scope, just as a transfer occurs in learning based on an already completed model.

*3.2. Learning Orientation Test Tools*

Many tools have been developed in the educational and psychological domains, including learning style inventory (LSI) [48], Gregorc's style delineator (GSD) [49], and

index of learning styles (ILS) [50]. However, we used the tool developed for the information literacy survey, conducted from October 2013 to January 2014, by the Japanese Ministry of Education, Culture, Sports, Science, and Technology, which was translated and used to suit the purpose of this study. The tool was selected considering that it was used at the national level, targeting elementary and middle school students in the field of computer science.

It would be safe to say that the validity of the learning orientation test tool was verified at the national level, but we re-tested its reliability for this study. It was analyzed as reliable with Cronbach's alpha ($\alpha$) at 0.912 for 13 items. There is no clear-cut criterion for Cronbach's $\alpha$, but it is said to be reliable if the value is 0.6 or higher; if it is lower than 0.6, the questionnaire should be discarded or revised [51]. As for the coefficients of factors, the survey method was 0.773, the analysis method was 0.750, and the thought organization and decision method was 0.862. Since the reflection method consists of one factor, the reliability coefficient, which measures the consistency between factors, was not measured. Table 3 shows the reliability coefficient for learning orientation test tools.

**Table 3.** Learning orientation test tools.

| Factor | Elements | Cronbach's $\alpha$ |
|---|---|---|
| Research method | LearningType1. When I find something I don't understand, I try to research it in several ways, including looking up a dictionary or doing an internet search. | 0.773 |
| | LearningType2. I try to find out something I don't know by reading books (except textbooks, reference books, comic books, or magazines) | |
| | LearningType3. When I research, I try to collect as much as data I can to find out what I am looking for. | |
| Analysis method | LearningType4. When it is hard to understand by reading sentences, I try to express information through figures or tables. | 0.750 |
| | LearningType5. I compare collected data to find the commonalities and differences. | |
| | LearningType6. I think differently from others or try to think in my own way. | |
| Thought arrangement and decision-making methods | LearningType7. I think carefully about whether the story I heard or the data I collected is true or not. | 0.862 |
| | LearningType8. When there are different opinions, I listen to both sides carefully and decide by myself which one is right. | |
| | LearningType9. I create a new thing or add my own thoughts based on my research. | |
| | LearningType10. When I talk about my thoughts or opinions in front of my friends, I try to organize what I want to say. | |
| | LearningType11. When I find a problem, I try to think of a solution first and then make a suggestion. | |
| | LearningType12. I try to work with my friend to study or teach each other. | |
| Reflection method | LearningType13. I try to evaluate what I liked or did not like after learning or experiencing. | - |

### 3.3. Data Collection and Sampling

Korea developed a plan to make SW education compulsory at the elementary school level, and the Ministry of Education conducted an SW education research school (elementary/middle/high school) from 2015 to 2018 to establish the curriculum. To assess the characteristics influencing the modeling, samples were collected from SW education research schools in the following steps:

Step 1: The SW competency and learning orientation tests were conducted for two months, from April to May 2017, targeting 3486 students in the 5th and 6th grades of 19 SW education research schools (elementary schools).

Step 2: The data on 141 students who did not participate in both SW competency and learning orientation tests were excluded.

Step 3: A total of 3345 people who participated in both the SW competency test tool and the learning orientation questionnaire were divided by gender.

Step 4: To clearly identify the factors of the upper and lower groups, samples excluding within 1 standard deviation (approximately 68%) were extracted centered on the normal distribution mean of the modeling standard scores. The standard score is a dimensionless number that shows the position occupied on the standard deviation of the normal distribution. The number of samples extracted for the experiment is approximately 32% of the upper and lower groups, and there are 756 male students and 688 female students.

### 3.4. Analysis Methods

In this study, statistical techniques and machine learning techniques were used to analyze gender differences in learning propensity. To analyze the characteristics of the independent variable with the highest influence on modeling, we used the feature selection technique of Mallow's $C_p$, Akaike information criterion (AIC), Bayes information criterion (BIC), and adjusted $R^2$ [47]. Independent variables with a high influence on modeling were selected from the four methods and applied to XGBOOST [52]. Finally, the feature importance of the independent variable was calculated in the model with the highest accuracy of XGBOOST. The description of the number and characteristics of data generated using the analysis method is presented in Table 4.

**Table 4.** Data description by analysis method.

| Div. | Feature Selection | XGBOOST | |
|---|---|---|---|
| No. of Data | Male: 756<br>Female: 688 | Training Data<br>Male: 604<br>Female: 552 | Test Data<br>Male: 152<br>Female: 136 |
| Data Characteristics | Dependent variable:<br>Modeling<br>Independent variable:<br>3 SW competencies<br>excluding<br>modeling, 13 learning types | Dependent variable: Modeling<br>Independent variable:<br>High-influence characteristics<br>extracted from the feature<br>selection technique | |

Our rationale for using a regression model-based feature selection, a statistical technique, and XGBOOST, which is a data-mining technique, is that XGBOOST can complement and verify the feature selection results. That is, feature selection has the advantage that it can easily identify independent variables that influence the dependent variable and increase stability by removing the independent variable with low influence. However, a limitation to the feature selection method is that it should be determined whether there is an interaction between variables or whether it is suitable for a nonlinear model. To overcome this limitation, XGBOOST, an ensemble algorithm that solves the interaction and overfitting problems, was utilized.

### 3.4.1. Feature Selection

Feature selection can find the subset of the most useful features among raw data for improving classification or model accuracy and is used to quantify the influence among features [53]. Moreover, it is used to remove redundant or irrelevant features in machine learning so as to create predictive models and find a combination of features related to the class to be predicted [47,54].

In this study, in addition to Adjusted $R^2$, which is generally used in linear regression models, three techniques were applied to extract features influential in modeling. Typical methods of feature selection can be summarized as follows:

1. Mallow's $\mathbf{C_p}$ technique: $\mathbf{C_p}$ is the standardized sum of squared error (SSE) estimator of the data. $\mathbf{C_p}$ is the number of independent variables selected for model prediction; $n$ is the number of samples; and $\hat{\sigma}^2$ is the residual sum of squares for all features. The lower the $\mathbf{C_p}$ is, the better the model; the formula is presented below.

$$\mathbf{C_p} = \frac{\text{SSE}_p}{\hat{\sigma}^2} - (2p - n) \tag{1}$$

2. AIC technique: AIC is calculated using Equation (2) for the number of features $p$ selected for model prediction. As $p$ increases, compared to $\mathbf{C_p}$, the penalty becomes bigger, and when the sample $n$ is different, it becomes inaccurate. The lower the AIC, the better the model.

$$\text{AIC}_p = n\ln\left(\frac{\text{SSE}_p}{n}\right) + 2(p + 1) \tag{2}$$

3. BIC technique: The BIC for $p$ independent variables is shown in Equation (3). It is similar to AIC, but by modifying the last term, the disadvantage of AIC, which becomes inaccurate when sample $n$ becomes large, is complemented. As with AIC, it is better when the BIC becomes lower.

$$\text{BIC}_p = n\ln\left(\frac{\text{SSE}_p}{n}\right) + (p + 1)\ln n \tag{3}$$

4. Adjusted $R^2$ technique: As the number of features increases, the number of samples also increases. At this point, the explanatory power $R^2$ unconditionally increases. Adjusted $R^2$ is the applied penalty, as the number of samples ($n$) is reflected, as shown in Equation (4). It has a more meaningful value by receiving a penalty according to the number of samples and becomes a better model as adjusted $R^2$ becomes higher.

$$\text{Adjusted } R^2 = 1 - \left(\frac{\text{SSE}/(n - n - 1)}{\text{SST}/(n - 1)}\right) \tag{4}$$

### 3.4.2. XGBOOST

XGBoost (extreme gradient boosting) was developed to solve the interaction and overfitting problems in linear regression or tree-based models and improve the stability and training speed of large datasets. It is a decision tree-based algorithm that uses a boosting technique, with a low error value, by connecting several classification and regression trees (CART) [52].

Boosting is a sequential process where errors are corrected by using the information obtained from the previous tree to generate the next tree. In general gradient boosting, if a negative loss occurs during tree pruning, the process is stopped. However, XGBoost proceeds to the maximum depth of the tree specified as a parameter during the model execution process. If the improvement in the loss function does not reach a certain level, the tree pruning process is performed in the reverse direction [52].

Equation (5) is the objective function of XGBoost, where $L$ is a loss function, $k$ is the number of trees, and $\Omega$ is the complexity of the tree. This balances the bias and variance

trade-offs. The information gain for the node can be calculated while the objective function goes through the process of convergence.

$$\text{Obj} = \sum_i^n (y_i, \hat{y}_i) + \sum_{k=1}^k \Omega(f_k) \tag{5}$$

The information acquisition amount, gain, for a specific depth of the tree is shown in Equation (6) below.

$$\text{Gain} = \frac{G_L^2}{H_L + \lambda} + \frac{G_R^2}{H_R + \lambda} - \frac{(G_L^2 + G_R^2)^2}{H_L + H_R + \lambda} - \lambda \tag{6}$$

L is the left child score, and *R* is the right-side children score. The tree with the highest score is combined by repeating the process of creating the tree with the maximum gain value. Through this process, the optimal classification model and important features that affect the classification model can be extracted.

## 4. Results

### 4.1. Average Analysis Result of Learning Orientation by Factor

Table 5 shows the result of comparing the factors of learning orientations—the study method, analysis method, thought organization and decision method, and reflection method—by gender into the high-achieving and low-achieving groups.

**Table 5.** Learning orientation analysis result by gender and achievement levels.

| Factor | Male Students | | | Female Students | | |
|---|---|---|---|---|---|---|
| | High-Achieving Group | Low-Achieving Group | *t*-Value | High-Achieving Group | Low-Achieving Group | *t*-Value |
| | Mean (Std.) | Mean (Std.) | | Mean (Std.) | Mean (Std.) | |
| Research Method | 3.69 (0.88) | 3.31 (0.84) | 5.657 *** | 3.80 (0.79) | 3.41 (0.93) | 5.903 *** |
| Analysis Method | 3.31 (0.89) | 3.15 (0.85) | 2.294 * | 3.35 (0.80) | 3.16 (0.88) | 2.830 ** |
| Thought Organization and Decision-Making Method | 3.67 (0.73) | 3.40 (0.79) | 4.720 *** | 3.78 (0.69) | 3.38 (0.79) | 6.634 *** |
| Reflection Method | 3.29 (1.11) | 3.27 (1.10) | 0.182 | 3.44 (0.97) | 3.26 (1.03) | 2.294 * |

* $p < 0.05$, ** $p < 0.01$, *** $p < 0.001$.

Male students showed a bigger difference between the high-achieving group and the low-achieving group by using the research method and thought organization and decision methods. In the survey method, the male students in the high-achieving group were 3.68 (0.88), indicating that they were more active than the male students in the low-achieving group, at 3.31 (0.84), with a statistically significant difference. Regarding the thought organization and decision method, the difference between the high-achieving group, at 3.67 (0.73), and the low-achieving group, at 3.40 (0.79), was statistically significant. However, no significant difference was found between groups for the reflection method. This implies that although the analysis and reflection methods may also be crucial factors, it is necessary to provide a teaching and learning method that can help the low-achieving group students with the use of the research method and thought organization and decision method.

Female students showed statistically significant differences in all factors regarding research method, analysis method, thought organization and decision method, and reflection method. Notably, for male students, the difference was bigger in the research method and thought organization and decision method. In the survey method, female students in the high-achieving group scored 3.80 (0.79), and those in the low-achieving group scored 3.41 (0.93). The difference between the high-achieving group and the low-achieving group was the biggest for the thought organization and decision method among female students at 3.78 (0.69) and 3.38 (0.79), respectively. The result can be interpreted in the same way as in the previous study that asserted that, for female students, it is necessary to conduct education faithful to the basics without ignoring all learning orientation factors [15].

### 4.2. Mean Analysis Result by the Factor of SW Competency and Learning Orientation

The results for each SW competency and learning orientation factor by gender are shown in Table 6.

**Table 6.** SW competency and learning orientation by gender.

| Div. | | Male Students | | Female Students | |
|---|---|---|---|---|---|
| | | High-Achieving Group | Low-Achieving Group | High-Achieving Group | Low-Achieving Group |
| | | Mean (Std.) | Mean (Std.) | Mean (Std.) | Mean (Std.) |
| SW Competency | Analysis | 73.31 (18.98) | 41.60 (20.17) | 72.14 (18.60) | 42.02 (21.07) |
| | Modeling | 88.80 (7.83) | 11.00 (7.91) | 88.44 (7.69) | 11.61 (7.68) |
| | Implementation | 43.96 (27.75) | 17.08 (20.53) | 37.89 (27.88) | 15.49 (19.87) |
| | Generalization | 72.23 (44.83) | 35.91 (48.07) | 69.38 (46.14) | 28.63 (45.30) |
| Learning Type | 1 | 3.88 (1.02) | 3.5 (1.08) | 4.06 (0.88) | 3.67 (1.09) |
| | 2 | 3.38 (1.11) | 3.05 (1.03) | 3.47 (1.04) | 3.21 (1.09) |
| | 3 | 3.81 (1.05) | 3.38 (1.02) | 3.87 (0.92) | 3.34 (1.03) |
| | 4 | 3.18 (1.14) | 3.1 (1.06) | 3.3 (1.04) | 3.15 (1.04) |
| | 5 | 3.13 (1.11) | 3 (1.05) | 3.2 (0.94) | 3.03 (1.08) |
| | 6 | 3.62 (1.01) | 3.36 (1.05) | 3.55 (0.98) | 3.29 (1.06) |
| | 7 | 3.67 (0.97) | 3.41 (1.05) | 3.7 (0.9) | 3.41 (0.96) |
| | 8 | 3.76 (0.96) | 3.46 (1.06) | 3.82 (0.87) | 3.46 (0.98) |
| | 9 | 3.62 (0.98) | 3.4 (1) | 3.69 (0.91) | 3.32 (0.98) |
| | 10 | 3.63 (0.94) | 3.37 (1.02) | 3.85 (0.9) | 3.32 (1.06) |
| | 11 | 3.63 (0.95) | 3.32 (0.96) | 3.72 (0.91) | 3.29 (1.01) |
| | 12 | 3.7 (1) | 3.42 (1.02) | 3.93 (0.89) | 3.49 (1.01) |
| | 13 | 3.29 (1.11) | 3.27 (1.1) | 3.44 (0.97) | 3.26 (1.03) |

The result of a mean analysis by gender in elementary schools showed that the difference in the score of modeling related to algorithm design was the largest in the SW competency area. For male students, the high-achieving group was 88.80 (7.83), and the low-achieving group was 11.00 (7.91), with a difference of 77.80. As for female students, the high-achieving group scored 88.44 (7.69), while the low-achieving group scored 11.61 (7.68), with a difference of 76.83. In short, modeling, as a step to establish a problem-solving procedure before implementing a program based on the analysis, showed the biggest learning gap, regardless of gender.

In learning orientation, the average of the high-achieving group was higher in all items. For male students, the learning type showing the most difference was data collection

(LearningType3) at 0.43, followed by dictionary and internet-based surveys (LearningType1) and book-based surveys (LearningType1). In contrast, the difference was the smallest for evaluation (LearningType13) at 0.02.

Among female students, the difference between the high-achieving group and the low-achieving group was the biggest for thought organization (LearningType10) and data collection (LearningType3) at 0.53 each, followed by devising and proposing a solution (LearningType11) and post-learning evaluation (LearningType12). In contrast, restructuring into figures or tables (LearningType4) showed the smallest difference of 0.15.

### 4.3. Feature Selection Results Influencing Modeling

Figure 1 shows the number of features with the greatest influence on modeling extracted from the high-achieving group and the low-achieving group by gender using the regression model.

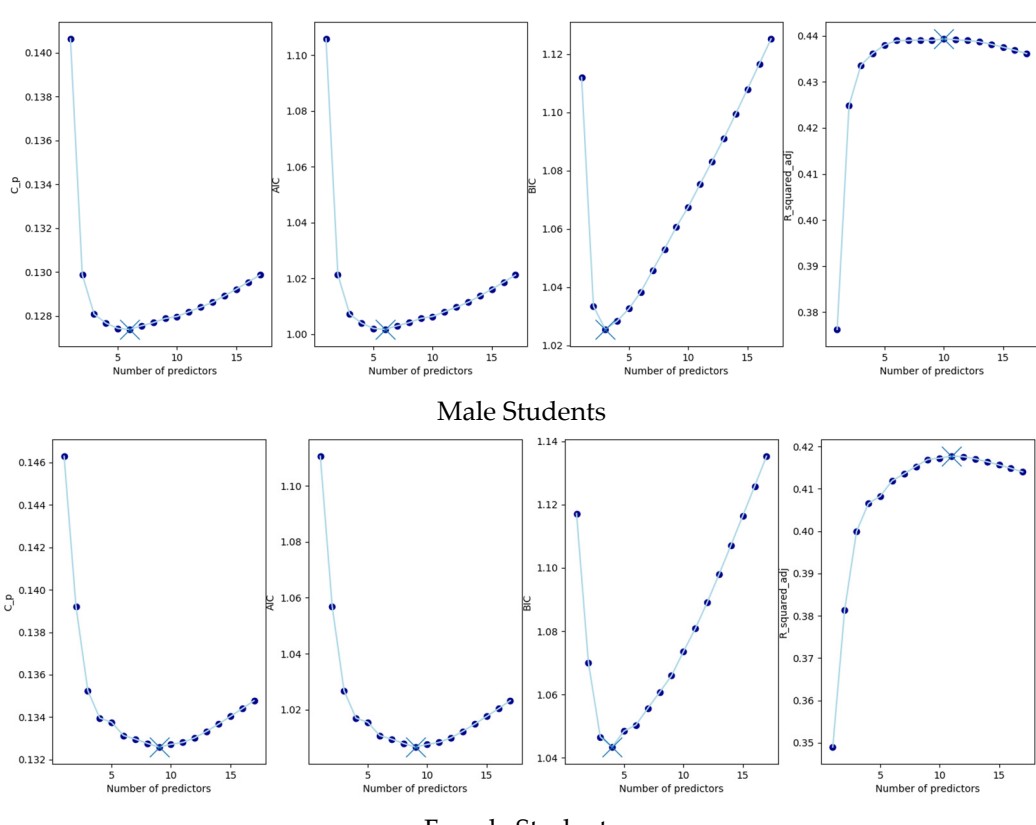

**Figure 1.** Feature selection (Mallow's $\mathbf{C}_p$, AIC, BIC, Adjusted $R^2$) results.

As for male students, Mallow's $\mathbf{C}_p$ and AIC showed that the model is the most suitable when there are six features. BIC had the lowest value when there were three features. Adjusted $R^2$ showed the highest value when there were 10 features. Female students exhibited the same nine features for Mallow's $\mathbf{C}_p$ and AIC. The model was the most suitable when the BIC had 4 features and adjusted $R^2$ had 11 features. Overall, more features were extracted from female students than from male students. For both male and female students, the least number of features were selected for the BIC, and the largest number for adjusted $R^2$.

The features with a great influence on modeling are shown in Table 7.

**Table 7.** Modeling: feature selection result.

| Div. | Male Students | | Female Students | |
|---|---|---|---|---|
| | **No. of Features** | **Order of Influence** | **No. of Features** | **Order of Influence** |
| Mallow's $\mathbf{C}_p$ | | | | Analysis<br>Generalization<br>Implementation<br>LearningType11<br>LearningType3<br>LearningType2<br>LearningType13<br>LearningType12<br>LearningType5 |
| AIC | 6 | Analysis<br>Implementation<br>Generalization<br>LearningType3<br>LearningType4<br>LearningType11 | 9 | |
| BIC | 3 | Analysis<br>Implementation<br>Generalization | 4 | Analysis<br>Generalization<br>Implementation<br>LearningType11 |
| Adjusted $R^2$ | 10 | Analysis<br>Implementation<br>Generalization<br>LearningType3<br>LearningType4<br>LearningType11<br>LearningType1<br>LearningType13<br>LearningType8<br>LearningType9 | 11 | Analysis<br>Generalization<br>Implementation<br>LearningType11<br>LearningType3<br>LearningType2<br>LearningType13<br>LearningType12<br>LearningType5<br>LearningType10<br>LearningType9 |

In Mallow's $\mathbf{C}_p$, AIC, BIC, and adjusted $R^2$, the three features of the biggest influence on modeling are analysis, implementation, and generalization for male students and analysis, generalization, and implementation for female students, in that order. This implies that, in the case of male students, the difference between the high-achieving group and the low-achieving group comes from the tendency to analyze and implement the problem and generalize through the process. However, for female students, modeling is influenced by the tendency to understand and the ability to implement in depth to a level that can be generalized based on the analyzed content.

For both male and female students, the same features were extracted in Mallow's $\mathbf{C}_p$ and AIC. In the modeling related to algorithm design, the order of influence was analysis, implementation, and data collection (LearningType3), restructuring into figures or tables (LearningType4), and devising and proposing a solution (LearningType11) for male students. For female students, the top five features were analysis, generalization, implementation, devising and proposing a solution (LearningType11), and data collection (LearningType3), all of which were selected for male students as well, although in a different order. Then, book-based learning (LearningType2), post-learning evaluation (LearningType12), collaboration (LearningType12), and feature comparison (LearningType5) followed. It can be interpreted as the between-group differences among female students being influenced by—as well as the top five features related to modeling—their willingness to find out something they do not know, cooperation with friends, reflection on learning, and comparison of the collected data.

The BIC includes analysis, implementation, and generalization for both male and female students, although in different orders. However, for female students, an additional feature was selected, that is, devising and proposing a solution (LearningType11).

As for adjusted $R^2$, common features, although in a different order, include analysis, implementation, generalization, data collection (LearningType3), and creation (Learning-

Type9). For male students, in addition to the shared features with female students, related to modeling, restructuring a problem into figures or tables, researching in various ways, and collecting opinions and judgments were selected as the features of learning orientation that influence between-group differences. For female students, in addition to common characteristics, book-based learning (LearningType2), post-learning evaluation (Learning-Type12), collaboration (LearningType12), feature comparison (LearningType5), and content organization (LearningType10) were selected as features with a high influence. This is the result of the selected features from Mallow's $\mathbf{C}_p$ and AIC, which are added by organizing thoughts and opinions (LearningType10) and creative thinking (LearningType9).

### 4.4. XGBOOST Result Applying Feature Selection

The features with a high influence on modeling were selected from Mallow's $\mathbf{C}_p$, AIC, BIC, and adjusted $R^2$ to apply to XGBOOST. The accuracy of XGBOOST for modeling by gender is shown in Table 8.

**Table 8.** Accuracy of XGBOOST.

| Gender | Div. (No. of Features) | Accuracy (%) |
|---|---|---|
| Male | Not Applied Feature Selection (16) | 76.97 |
| | Mallow's $\mathbf{C}_p$ (6) and AIC (6) | 80.26 |
| | BIC (3) | 76.97 |
| | Adjusted $R^2$ (10) | 78.29 |
| Female | Not Applied Feature Selection (16) | 81.16 |
| | Mallow's $\mathbf{C}_p$ (6) and AIC (9) | 81.88 |
| | BIC (4) | 79.71 |
| | Adjusted $R^2$ (11) | 81.56 |

Overall accuracy is higher when feature selection is applied than when in the "Not Applied Feature Selection" mode. For male students, the highest accuracy of XGBOOST is gained when a model is created using Mallow's $\mathbf{C}_p$ and AIC. The accuracy is 80.26%, 3.29% higher than when the BIC is applied, or no feature selection is applied.

For female students, the highest accuracy of 81.88% is gained for XGBOOST when a model is created after applying Mallow's $\mathbf{C}_p$ and AIC, as in the case of male students. The accuracy was the lowest when the influencing features from the BIC were applied to XGBOOST.

The difference in accuracy depending on the selected feature is related to the residual: The closer the mean to the sum of the residuals is to 0, the more ideally trained the model (fitted model) is, and the larger the mean to the sum of the residuals, the more inefficiently trained the model. If the selected features raised the accuracy of the prediction model, they can be interpreted as having a high influence on the predictive value.

Figure 2 shows the important features in XGBOOST (Mallow's $\mathbf{C}_p$ and AIC) with the highest accuracy by gender.

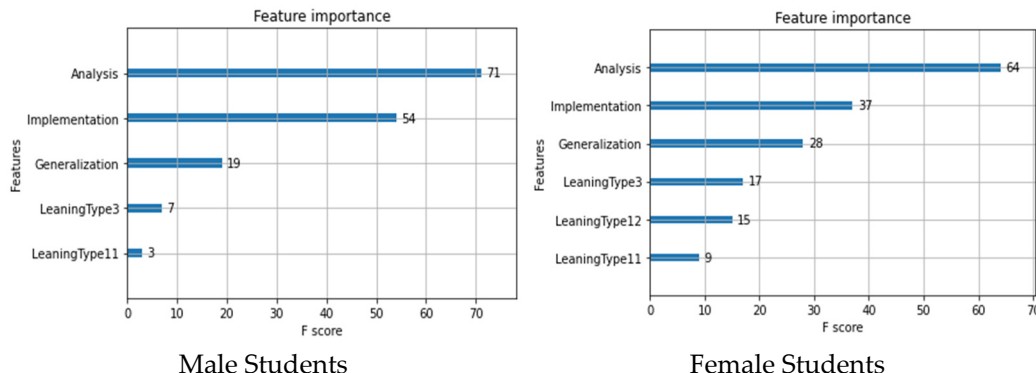

Male Students                                             Female Students

**Figure 2.** XGBOOST: Important features of male and female students.

In terms of the importance of features, regarding modeling, analysis 71 and implementation 54 had the biggest influence among other features on male students. Generalization was 19, data collection (LearningType3) was 7, and devising and proposing a solution (LearningType11) was 3. This means that the difference between the high-achieving group and the low-achieving group of male students mainly comes from the analysis and implementation of the problem. For female students, analysis (64) was the most influential; followed by implementation, generalization, and data collection (LearningType3); cooperation with friends (LearningType12); and devising and proposing a solution (LearningType11).

The features that male and female students shared were analysis, implementation, generalization, and data collection (LearningType3), and devising and proposing a solution (LearningType11). Regardless of gender, collection of relevant data and analysis, establishing a direction for problem solving, implementation into a program, and generalization, were the features influencing modeling. Besides the common features, cooperation with friends and teaching each other (LearningType12) were revealed as important features among female students.

## 5. Conclusions

An AI-based intelligent information society requires creative learners who can create new things. We cannot overestimate the importance of providing an educational environment tailored to the learning orientation of learners such that they can choose suitable learning strategies to enhance academic achievement [55]. In this sense, teachers' understanding of student characteristics is crucial in improving and developing students' learning abilities in the classroom.

This study focused on the stage of modeling in programming education, where elementary school students design algorithms. This study aimed to investigate the influence of learning orientation, as well as analysis, implementation, and generalization, on the stage of modeling by gender.

We focused on the modeling stage mainly for two reasons: First, it provides the basis of the design for automation in programming education. Second, learners engage in collaboration to solve problems when the issues learners face become increasingly complicated. In other words, modeling, as a way of communication, is closely related to the expression of one's idea or intention in various methods, along with expressions or emotions [56].

To examine the features influencing modeling in the high-achieving and low-achieving groups, by gender, the features with the high influences were extracted using basic statistical analysis and regression models. The extracted features were applied to the ensemble model, XGBOOST, to be verified again. In machine learning, the fact that Mallow's $C_p$ and AIC show high levels of accuracy, when the data of this study are applied, means that the trained model is more efficient than other feature selection techniques and, above all, that the generalization of the trained model through the most important features is well performed.

Based on the results of this study, we make the following suggestions.

First, learners, regardless of gender, show the biggest gap in modeling, where the difference in programming education occurs. Considering programming education as a process of analyzing, modeling, implementing, and generalizing a given problem, learners experiencing difficulty with modeling are also likely to find implementation and generalization difficult. For accuracy, it will be helpful to focus on programming, that is, the coding of algorithms through modeling. This must be accompanied by an understanding of the previous stage—modeling. Accordingly, learning about algorithms and modeling should be improved for all learners.

Second, the results of XGBOOST for male students, having high accuracy, found that data collection (LearningType3) and devising a solution (LearningType11) were important learning methods. The two methods can be considered integral in all educational areas, not only in programming education. However, since programming education tends to emphasize the functional aspects of creating an automated program, it may overlook the

two above-mentioned learning orientations. Therefore, more attention should be paid to the creation of a learning environment, where learners, during the process of programming, can collect and analyze the relevant data to derive a direction to solve a problem. For male students, it is necessary to create an environment where they can concentrate on deriving solutions from the collected data while reinforcing information search education in order to retrieve and utilize necessary information.

Third, the results of XGBOOST for female students, which had high accuracy, found that cooperation with friends (LearningType12) was an important learning orientation, contrary to their male peers, as well as data collection (LearningType3) and devising a solution (LearningType11). Given the previous finding that female students tend to be more interested in theories with higher achievement [15], it seems necessary to create an environment where they can apply the basics to practice by working on team projects [56]. Accordingly, it would be required to apply teaching–learning methods that enhance students' abilities to collaborate with friends in team projects.

As such, this study analyzed the effects of modeling according to the gender of elementary school students. In the modeling stage, no significant relationship was found between learning propensity according to gender, but collaboration was the most important factor in the case of female students. In addition, although it is often the case that only coding is emphasized in programming education, the modeling process can be seen as salient as the previous stage of coding. To increase the effectiveness of programming learning, it is judged that a more systematic analysis is needed not only on modeling, but also on preferences according to programming skills, problem solving methods, and the learning environment. In the future, research on the kind of achievement that each individual project and team project represents should be conducted.

This study's findings are expected to serve as basic data for teaching and learning design and operation for effective programming education.

**Author Contributions:** Methodology, H.W.; writing—original draft preparation, H.W.; writing—review and editing, H.W. and J.-M.K.; visualization, H.W.; validation, H.W. and J.-M.K.; funding acquisition, H.W. All authors have read and agreed to the published version of the manuscript.

**Funding:** This research was supported by the Korea National Open University Research Fund.

**Institutional Review Board Statement:** Not applicable.

**Informed Consent Statement:** Not applicable.

**Data Availability Statement:** Not applicable.

**Conflicts of Interest:** The authors declare no conflict of interest.

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
