# Peer review of "Impacts of Learning Orientation on the Modeling of Programming Using Feature Selection and XGBOOST: A Gender-Focused Analysis"

_applsci, doi:10.3390/app12104922_

Round 1
Reviewer 1 Report
The authors of the article show a high knowledge of the content and tools used in the research process. A pertinent clarification of the situation of women is shown to justify the approach of the investigation according to the gender variable.
Here are some revisions that could be used to improve the article:
• The summary could include a preview of the results achieved.
• Research adjusted to the proposed objectives, come to support the study developed, resulting in a good approach to the state of the matter. On the other hand, there is a lack of conceptualization of some key concepts, such as programming education.
• Regarding the instruments used, the methodological process followed is shown. In the first, it would have been interesting to have some more experts, especially in the initial phases. In relation to the second, more aspects of the process of adapting the instrument from one country to another could be indicated.
• In expert judgments, the content validity index and the Kappa index could have been calculated.
• The development and verification of the instruments could have been included in the Method section.
• High sample used. The sample selection method used could be specified.
• Pertinent results and with an adequate presentation. They could have been supported by other research.
• The suggestions presented in the conclusions are interesting and have pedagogical value. No limitations or proposals for future research are included.
• The bibliography used is appropriate to the research topic, although a greater number of references from the last five years should be included.
Author Response
The authors of the article show a high knowledge of the content and tools used in the research process. A pertinent clarification of the situation of women is shown to justify the approach of the investigation according to the gender variable.
Here are some revisions that could be used to improve the article:
• The summary could include a preview of the results achieved.
Line 11] Thank you for your valuable comments. Based on your suggestions, we have added research findings and key conclusions.
• Research adjusted to the proposed objectives, come to support the study developed, resulting in a good approach to the state of the matter. On the other hand, there is a lack of conceptualization of some key concepts, such as programming education.
Line 47] We have added a definition of programming to the introduction in response to reviewers' comments.
• Regarding the instruments used, the methodological process followed is shown. In the first, it would have been interesting to have some more experts, especially in the initial phases. In relation to the second, more aspects of the process of adapting the instrument from one country to another could be indicated. In expert judgments, the content validity index and the Kappa index could have been calculated.
Line 201] We fully agree with your opinion. In this study, SW Competency Assessment Tool and Learning Orientation Test Tools were used. It is thought that if more experts participated in the process of developing the SW Competency Assessment Tool, it would have been helpful to secure validity. However, in Korea, we tried to develop a systematic tool through face-to-face consultations with a small number of professors and teachers who are interested in research in the field of SW education. This researcher also regrets that the expert's content validity index or kappa index was not reflected. When conducting other research in the future, we will design the study to reflect the content. In the case of the learning propensity test tool, it can be said that the validity of the tool developed by the Ministry of Education, Culture, Sports, Science and Technology has been secured. Reliability verification of the tool was also performed to ensure feasibility. Although it was stated that the tool was modified and supplemented in this study, it was revised as translated and applied because there was a possibility of misunderstanding. Thank you again for your valuable comments.
• The development and verification of the instruments could have been included in the Method section.
Line 144] We have modified the names of Chapter 3 as a method according to your suggestion. In addition, we have placed the contents of Chapters 3 and 4 together in the same section.
• High sample used. The sample selection method used could be specified.
Line 233] In response to your comments, we have added a description of the sample.
• Pertinent results and with an adequate presentation. They could have been supported by other research.
I'm not sure we understood the reviewer's comments clearly. I'm sorry, but I'll explain based on what I understand. This researcher is interested in analyzing educational data using artificial intelligence and data mining. The SW competency measurement tool and learning propensity test tool used in this study are new tools that have not been utilized in other studies. The learning propensity test tool was used in the information literacy test of the Japanese Ministry of Education, Culture, Sports, Science and Technology, but analysis was not performed according to the characteristics of the learner. In other words, there are no studies that can refer to the contents related to this study. The research method uses two techniques. From what I have confirmed, in educational research in the field of data science, I have not been able to confirm research conducted by mixing Feature Selection and XGBOOST. This study is meaningful in that it used a new methodology for data analysis in the academic and educational fields. If you give us a specific opinion on the matter, we will be sure to reflect it.
• The suggestions presented in the conclusions are interesting and have pedagogical value. No limitations or proposals for future research are included.
Line 522] Reflecting the opinions of reviewers, the academic significance of this study and directions for future research are presented from an educational point of view.
• The bibliography used is appropriate to the research topic, although a greater number of references from the last five years should be included.
Line 544] Thank you for your valuable comments. References within the last 5 years have been added and supplemented.
Reviewer 2 Report
Dear Authors:
The study you present is interesting and relevant. However, I have found
some limitations that need to be addressed:
- In the abstract the results and main conclusions should be made
explicit. In other words, the last part of the abstract should clearly
indicate the results obtained and the main conclusions of the research.
- The section "Experimental Methods" should be renamed to "Methods". At
the beginning of this section, the design used in the research should be
explained and referenced. In this sense, the nature of the study should
be indicated, as well as the methodological design used, indicating the
corresponding theoretical references that provide the possibility of
contrasting and replicating the research. In this section it is also
important to explain in detail the validation process followed for the
instruments used, in order to clarify the reliability of the instruments
used.
- Before the conclusions, a discussion of results section should be
included, where the results obtained here are compared with those
published in previous research. In this sense, the authors should carry
out an exhaustive bibliographic review to locate the studies carried out
in this line of research and then compare their results with those
obtained here. The discussion should be detailed, since there is no
mention of this issue in the manuscript. Therefore, all aspects related
to the results obtained should be taken into account in order to provide
more clarity and light to those obtained here.
- At the end of the conclusions section, the limitations encountered in
the course of the research should be clearly indicated, so that the
results would be more contextualized.
- The bibliographic references used are adequate, but more attention
should be paid to the most current and cutting-edge bibliography in this
line of research.
Author Response
The study you present is interesting and relevant. However, I have found
some limitations that need to be addressed:
- In the abstract the results and main conclusions should be made
explicit. In other words, the last part of the abstract should clearly
indicate the results obtained and the main conclusions of the research.
Line 11] Thank you for your valuable comments. Based on your suggestions, we have added research findings and key conclusions.
- The section "Experimental Methods" should be renamed to "Methods". At the beginning of this section, the design used in the research should be explained and referenced. In this sense, the nature of the study should be indicated, as well as the methodological design used, indicating the corresponding theoretical references that provide the possibility of contrasting and replicating the research. In this section it is also important to explain in detail the validation process followed for the instruments used, in order to clarify the reliability of the instruments used.
Line 144] We've changed the name of Section 4 to “Methods” to reflect our reviewers' comments. The tool development and validation in Chapter 3 were included in the research method. We have also added a brief description of the research methods used at the beginning of this section, along with references.
- Before the conclusions, a discussion of results section should be
included, where the results obtained here are compared with those
published in previous research. In this sense, the authors should carry
out an exhaustive bibliographic review to locate the studies carried out in this line of research and then compare their results with those
obtained here. The discussion should be detailed, since there is no
mention of this issue in the manuscript. Therefore, all aspects related
to the results obtained should be taken into account in order to provide more clarity and light to those obtained here.
We fully agree with your opinion. In the case of existing research, research on programs, which are programming products, is the main focus. On the other hand, this study focused on the modeling process and learning tendency to design algorithms in the programming process. In other words, it is difficult to directly compare research results because programming and modeling, which are outputs in programming education, are considered different steps in the procedure.
- At the end of the conclusions section, the limitations encountered in
the course of the research should be clearly indicated, so that the
results would be more contextualized.
Line 522] We reflected the opinions of reviewers and presented the limitations of this study and directions for future research.
- The bibliographic references used are adequate, but more attention
should be paid to the most current and cutting-edge bibliography in this line of research.
Line 544] Thank you for your valuable comments. References within the last 5 years have been added and supplemented.
Reviewer 3 Report
The paper is well written and sound. There is some room for improvement and I will list my observations below. I am not an expert on the methods you have used in the sections 4.2.1 - 5. - so I have not assessed these sections of your manuscript.
1) English language. It is mostly fine, but there are some misspellings, and some grammar errors (articles, singular/plural (e.g. on the row 119)). It needs to be revised to achieve an immaculate submission.
2) Fonts & format. In some places the font changes and is not according to the journal standard. For example, on the rows 29, 82, 134... Please check this fully.
3) Scientific buildup of your work. The details will follow, but shortly, in general - (a) you have left some concepts (that you are using) undescribed; (b) you have omitted some necessary references; and (c) you have left out some small (but necessary) sections. All of these problems can be corrected quite easily.
===
Description of concepts. In 2.1 you talk about GEFT, programming design and coding (the rows 75-77). In order for the readers to grasp the context, please open/describe these context, with proper referencing, and describe their correlation. The construct "programming design" is mentioned only once in the paper - if you want to include it, state its significance (in relation to GEFT). You refer to correlation between GEFT and programming design - but these concepts belong into different categories and can not be correlated - perhaps you wanted to refer to correlation of some GEFT results and level of student programming design?
On the row 77 you refer to some stages of the programming process - please open the concept programming process and list its stages.
On the row 93 you state that learning orientation is associated with academic achievements. Please describe in some details, how (what aspects are associated in what way).
On the rows 114-119 you speak about attitude as if it was something that can be linearly rated. Attitude in itself is not something you can measure in such a way. However, there can be positive attitudes, negative, bad, good attitudes, and the prevalence of those can be measured. You either need to choose another word for what you want to express or, alternatively, you would need to rephrase the sentences on the rows 114-119.
On the rows 147-148, please describe the conducted literature review shortly (and add a reference if it is already published).
On the rows 161-162 you speak about a test tool. It only has 17 questions in it, please list these questions - or, alternatively, describe 4 question categories in more detail (either as a separate list, or in Table 2).
On the row 184 you refer to a study. Please add a literature reference to the published paper here.
On the row 230 you are making a claim about XGBOOST. Please add a reference that supports your claim. Similarly, on the rows 236-237 you are making another claim that would need explanation (how does it solve...) and a reference if you are relying on the previous literature.
The section "6. Conclusion". On the rows 487-514 you bring out three suggestions. Please connect your suggestions to existing literature by adding comparative references to similar studies.
Finally, you are missing the section "Limitations and Future Work".
You would need to describe the limitations or potential shortcomings of your paper. You would need to describe what other studies in this area should be done and how could these be built on your results.
Author Response
The paper is well written and sound. There is some room for improvement and I will list my observations below. I am not an expert on the methods you have used in the sections 4.2.1 - 5. - so I have not assessed these sections of your manuscript.
1) English language. It is mostly fine, but there are some misspellings, and some grammar errors (articles, singular/plural (e.g. on the row 119)). It needs to be revised to achieve an immaculate submission.
We've re-checked the spelling and grammar based on your comments. thank you.
2) Fonts & format. In some places the font changes and is not according to the journal standard. For example, on the rows 29, 82, 134... Please check this fully.
Based on your comments, we modified the format to conform to journal standards. Thank you for your valuable comments.
3) Scientific buildup of your work. The details will follow, but shortly, in general - (a) you have left some concepts (that you are using) undescribed; (b) you have omitted some necessary references; and (c) you have left out some small (but necessary) sections. All of these problems can be corrected quite easily.
Description of concepts. In 2.1 you talk about GEFT, programming design and coding (the rows 75-77). In order for the readers to grasp the context, please open/describe these context, with proper referencing, and describe their correlation. The construct "programming design" is mentioned only once in the paper - if you want to include it, state its significance (in relation to GEFT). You refer to correlation between GEFT and programming design - but these concepts belong into different categories and can not be correlated - perhaps you wanted to refer to correlation of some GEFT results and level of student programming design?
On the row 77 you refer to some stages of the programming process - please open the concept programming process and list its stages.
Line 201] Thanks for your comments. In understanding this paper, the content has been removed because there is a possibility of misunderstanding.
On the row 93 you state that learning orientation is associated with academic achievements. Please describe in some details, how (what aspects are associated in what way).
Line 96] We have improved the content based on your comments.
On the rows 114-119 you speak about attitude as if it was something that can be linearly rated. Attitude in itself is not something you can measure in such a way. However, there can be positive attitudes, negative, bad, good attitudes, and the prevalence of those can be measured. You either need to choose another word for what you want to express or, alternatively, you would need to rephrase the sentences on the rows 114-119.
Line 123] Attitude is a difficult factor to measure and we totally agree with you. With reference to the paper, an explanation of the attitude has been added.
On the rows 147-148, please describe the conducted literature review shortly (and add a reference if it is already published).
Line 158] In response to reviewers' comments, we have provided reference links. In addition, the main points of the literature review are further explained.
On the rows 161-162 you speak about a test tool. It only has 17 questions in it, please list these questions - or, alternatively, describe 4 question categories in more detail (either as a separate list, or in Table 2).
Line 180] Reflecting your opinion. Detailed factors for 17 items were additionally explained.
On the row 184 you refer to a study. Please add a literature reference to the published paper here.
Line 201] In response to your comments, we have added references to published articles.
On the row 230 you are making a claim about XGBOOST. Please add a reference that supports your claim. Similarly, on the rows 236-237 you are making another claim that would need explanation (how does it solve...) and a reference if you are relying on the previous literature.
Line 263] In response to your comments, we have added references to published articles.
The section "6. Conclusion". On the rows 487-514 you bring out three suggestions. Please connect your suggestions to existing literature by adding comparative references to similar studies.
Line 514] We fully agree with your opinion. In the case of existing research, research on programs, which are programming products, is the main focus. On the other hand, this study focused on the modeling process and learning tendency to design algorithms in the programming process. In other words, it is difficult to directly compare research results because programming and modeling, which are outputs in programming education, are considered different steps in the procedure.
We have added references to published articles to reflect your suggestions as best as possible. In addition, the limitations of this study and directions for future research are presented.
Finally, you are missing the section "Limitations and Future Work".
You would need to describe the limitations or potential shortcomings of your paper. You would need to describe what other studies in this area should be done and how could these be built on your results.
Line 522] We reflected the opinions of reviewers and presented the limitations of this study and directions for future research.
Round 2
Reviewer 2 Report
The authors have taken my comments into account and have corrected certain limitations. However, the following indications are still missing:
- The main conclusions have not been made explicit in the abstract.
- Neither the nature of the study (quantitative, qualitative, mixed...) nor the type of design used in the research has been indicated at the beginning of the methods section. An overview of these issues should always appear in the first line of the methods section, before any explanation.
I hope you can remedy these two minor issues.
Author Response
Thank you very much for the reviewer's valuable comments.
We have revised abstract(Line 11) and the first line of the methods section(Line 152) to reflect your comments.